# Community-based health insurance beneficiaries' satisfaction on laboratory services and associated factors in selected public hospitals in Jimma Zone, Oromia Region, Southwest, Ethiopia

Nigusu Getachew[1]*, Mujahid Girma[2], Zewudineh Sahilemariam[2], Temesgen Kabeta[1], Amit Arora[3,4,5,6,7]

1 Department of Health Policy and Management, Faculty of Public Health, Institute of Health, Jimma University, Jimma, Ethiopia, 2 Department of Medical Laboratory Sciences Institute of Health Science, Jimma University, Jimma, Ethiopia, 3 School of Health Sciences, Western Sydney University, Penrith, New South Wales, Australia, 4 Translational Health Research Institute, Western Sydney University, Campbelltown, New South Wales, Australia, 5 Health Equity Laboratory, Campbelltown, New South Wales, Australia, 6 Discipline of Child and Adolescent Health, The Children's Hospital at Westmead Clinical School, Faculty of Medicine and Health, The University of Sydney, Westmead, New South Wales, Australia, 7 Oral Health Services, Sydney Local Health District and Sydney Dental Hospital, NSW Health, Surry Hills, New South Wales, Australia

* nigusgetachew45@gmail.com

## Abstract

### Background

The community-based health insurance (CBHI) scheme is a growing initiative aimed at enhancing healthcare access for the most impoverished members of the community. The Ethiopian CBHI scheme aims to enhance access to essential healthcare services, including medical laboratory services, for the poorest members of the community, but there is limited evidence on satisfaction levels. The aim of this study was to assess the satisfaction level of CBHI beneficiaries with laboratory services and their associated factors among selected public hospitals in Jimma Zone, Oromia Region, Ethiopia.

### Methods

A facility-based cross-sectional study was conducted on selected public hospitals in the Jimma Zone from September to October 2023. A total of 421 CBHI beneficiaries were enrolled in the study using a convenient sampling technique, and interviewers administered structured questionnaires to collect data. Data were entered into Epi-data and analyzed using the Statistical Package of Social Sciences version 25. Descriptive analysis was used to summarize independent variables; bivariate and multivariable logistic regression analyses were done to test the association between independent and dependent variables; and statistical significance was declared at P<0.05.

**Data Availability Statement:** All relevant data are within the manuscript and its Supporting information files.

**Funding:** The author(s) received no specific funding for this work.

**Competing interests:** The authors have declared that no competing interests exist.

## Results

More than half (55.8%) of the 419 study participants were female. Above half, 57.5% of the respondents were satisfied by the clinical laboratory services at public hospitals in Jimma Zone. Components with a higher satisfaction rate were providers' professional appearances (98.3%), procedures for specimen collection (87.6%), and availability of entertainment facilities at the waiting area (67.8%). On the contrary, longer waiting times to receive the test results (76.6%), inefficiency of the reception area (74.7%), and the inability of professionals to explain diagnostic procedures (58.0%) were associated with higher rates of dissatisfaction. Educational status and the number of hospital visits were found to have a statistically significant association with level of satisfaction with laboratory services.

## Conclusions

CBHI beneficiaries' satisfaction with laboratory service was at a moderate level in Jimma Zone public hospitals. Therefore, attention should be given to continuous monitoring of patients' satisfaction with services, improving reception areas, and practicing routine explanations about the purposes and procedures during specimen collection to improve the beneficiaries' satisfaction level with clinical laboratory services.

## Background

Universal health coverage (UHC) has been a global health priority, aiming to ensure that all individuals have access to quality health services without facing financial hardship. It includes essential health services like health promotion, prevention, treatment, rehabilitation, and palliative care. The World Health Organization(WHO) advocates for UHC worldwide, focusing on global health diffusion and progress tracking, but concerns have been raised about overemphasizing medical interventions and neglecting other essential aspects of healthcare delivery [1–3]. UHC in developing countries is challenging due to over-reliance on out-of-pocket spending (OOPS). Ethiopia's OOPS share is 32.3%, exceeding the global target of 15–20% in low-income countries. To address this, Ethiopia is implementing community-based health insurance schemes [4–8].

Healthcare policymakers worldwide face challenges in financing healthcare systems. The WHO recommends a risk-pooling prepayment approach, a strategy where individuals contribute to a common fund that is used to cover healthcare costs for the entire group [9–12]. CBHI has emerged as an alternative to user fees, which can be a barrier to accessing necessary healthcare. CBHI is a voluntary health insurance scheme funded through a variety of sources, including central government contributions, private sector support, donor organizations, NGOs, and investment dividends [13–15].

CBHI offers several potential benefits. Studies have shown that CBHI programs can increase healthcare access, especially for vulnerable populations [10]. Additionally, CBHI can reduce the financial burden on individuals and families and potentially improve health outcomes due to earlier treatment seeking behavior. However, CBHI faces challenges such as enrollment rates, payment capacity, and long-term financial sustainability [13, 16–20].

CBHI schemes improve healthcare access in low- and middle-income countries by providing financial protection and reducing OOPS and social capital. These schemes encourage individuals to seek medical care, leading to earlier diagnosis and treatment. However, challenges persist in ensuring equitable access to healthcare services, especially among vulnerable

populations. CBHI plays a crucial role in shaping healthcare utilization patterns and social capital, but barriers to implementation, uptake, and sustainability are critical factors affecting their effectiveness. A comprehensive review is needed to assess CBHI's impact on health outcomes and UHC goals [9, 10, 11, 21–25]. Client satisfaction refers to a person's overall perception of a healthcare experience, and poor service provision can lead to financial losses and jeopardize the sustainability of the healthcare insurance industry [26–28].

Client satisfaction with healthcare services under CBHI is significantly higher than without CBHI, largely due to beneficiary expectations. Studies across various developing countries, including Ethiopia, have identified Key predictors include wait time, staff friendliness, the consultation process, socio-demographic factors, the availability of diagnostics and drugs, and the facility environment. However, concerns about providers denying entitlements and charging additional fees raise questions about potential scams. Research on CBHI in low-income settings highlights increased hospitalization rates, but low enrollment threatens scheme sustainability. Globally, over 7.3 billion people lack access to essential health services, leading to poverty and high OOPS. Government investment is needed to improve drug availability and address worker complaints [22, 29–40]. A pilot CBHI scheme, launched in 2011, has improved health outcomes and empowered women. However, challenges like provider readiness, drug shortages, moral hazards, equipment breakdowns, staff shortages, and complaint collection mechanisms hinder service quality [41–44].

The quality of laboratory service is crucial for healthcare delivery, but a mismatch between client expectations and service can lead to dissatisfaction. Client satisfaction is often neglected during health insurance schemes, influenced by staff professionalism, information collection, waiting times, availability, cleanliness, room location, availability of essential resources like medications and laboratory services, and latrine accessibility [45, 46].

Client satisfaction is crucial for the CBHI program, as it increases beneficiary renewal rates and attracts new members. Surveys provide feedback on care quality, availability, and continuity, ensuring the program's sustainability [47, 48]. Compared to the general population, CBHI members represent a specific group with distinct experiences within the healthcare system. They actively participate by contributing premiums and utilizing CBHI services. Understanding their satisfaction with specific aspects, like laboratory services, provides valuable insights for program improvement. According to our knowledge and access until the writing of this paper, there has been no research conducted directly on the issues in the study setting; therefore, this research explores the satisfaction of CBHI beneficiaries with laboratory services in selected public hospitals in Jimma Zone, Oromia Region, Southwest Ethiopia, aiming to improve the quality and effectiveness of CBHI programs.

## Methods

### Study design, setting, and period

A facility-based cross-sectional study was conducted in Jimma Zone hospitals from September to October 2023. There are about 377,913 payer households of CBHI beneficiaries and about 67,616 indigents. There are 9 public hospitals, including Jimma University Medical Center, 3 primary private hospitals, 120 functional health centers, and 561 health posts in Jimma Zone. The study focused on six hospitals offering community-based health insurance (CBHI) services.

### Population

The source and study population were all CBHI beneficiaries and CBHI beneficiaries who attended the selected hospitals for laboratory services during the study period, respectively.

**Eligibility criteria.** The study included CBHI beneficiaries who attended selected hospitals for laboratory services during the study period. Participants who are unable to hear or speak, and who did not utilize laboratory services during their hospital visit were excluded from the study.

## Sample size determination and sampling procedure

The sample size for this study was determined using a single population formula, assuming a 95% confidence interval, a desired precision of 5%, and a 10% non-response rate. Based on a previous facility-based cross-sectional study conducted in Addis Ababa, Ethiopia [49], a 53% proportion of satisfaction with laboratory services among CBHI beneficiaries was anticipated. This resulted in a minimum sample size of 383, which was adjusted to 421 to account for the potential non-response rate. Simple random sampling with a lottery method was used to select fifty percent (50%) of the hospitals (three out of the six hospitals) in the Jimma Zone that offered CBHI services, as confirmed by the Jimma Zone CBHI coordinating office. A convenient sampling technique was used to select the study participants until our total sample size was acquired. The total number of CBHI beneficiaries was 88,449 in three randomly selected hospitals: Agaro, Seka Chokorsa, and Shenen Gibe. The sample size was proportionally allocated to each hospital based on its total number of enrolled CBHI beneficiaries.

## Study variables

The dependent variable was the satisfaction level of CBHI beneficiaries, and the independent variables were socio-demographics, laboratory service provision-related determinants of CBHI beneficiaries' satisfaction, and socio-economic characteristics of the CBHI beneficiaries like household wealth index, household family size, and occupation.

## Operational definitions

**Satisfaction level.** Overall patients' satisfaction level was calculated by taking the mean satisfaction score of all the 20 questions used to assess satisfaction level and classified into two categories as follows.

**Dissatisfied.** Patients whose score is below the mean satisfaction score of all the 20 questions used to assess satisfaction level.

**Satisfied.** Patients whose score is above the mean satisfaction score of all the 20 questions used to assess satisfaction level.

## Data collection instrument

The data were collected using a face-to-face interview with CBHI beneficiaries by trained laboratory personnel with a pre-tested questionnaire. Data collectors were trained with the objective of standardizing the data collection instrument and providing them with the basic skill of extracting the data. The questionnaires were arranged after revising different related journals outside and inside the country, referring from global to local. In addition, study participants were asked to rate each aspect of their laboratory services on five-point scales (very dissatisfied, dissatisfied, neutral, satisfied, and very satisfied). A 5-point Likert scale rating of very dissatisfied (1 point), dissatisfied (2 points), neutral (3 points), satisfied (4 points), and very satisfied (5 points) was used. To calculate the level of patient satisfaction with different laboratory services, mean satisfaction was calculated, and respondents whose total score was mean or above were classified as satisfied, while those scoring below the mean score were categorized as dissatisfied. The overall rate of satisfaction by Likert scales was calculated as (number of very

satisfied rating ×5) + (number of satisfied rating ×4) + (number of neutral rating ×3) + (number of dissatisfied rating ×2) + (number of very dissatisfied rating ×1) divided by the total number of ratings (1–5) for the general laboratory services. The internal consistency of the items was evaluated using Cronbach's alpha, with items with a Cronbach's alpha greater than 0.6 being included in the final analysis. Satisfaction with the facility has seven items (Cronbach's alpha = 0.62), satisfaction with laboratory professionals service provision and ethical characteristics has six items (Cronbach's alpha = 0.69) and satisfaction with the availability of tests and service-related issues has seven items (Cronbach's alpha = 0.62).

### Data processing and analysis

The completed questionnaires were coded and entered into EPI-DATA for data cleaning and then transferred to SPSS version 25 for further analysis. The goodness of fit was checked with the Hosmer–Lemeshow test (p = 0.35). Multicollinearity was checked by examining the variance inflation factor. Descriptive statistics were performed to summarize the data, with results presented as percentages for categorical variables and means with standard deviations for continuous variables. To identify factors associated with satisfaction with laboratory services, bivariate logistic regression analyses were initially conducted. Variables with a p-value less than 0.25 in the bivariate analysis were then included in a multivariable logistic regression model to control for potential confounding factors. The study used odds ratios (OR) to assess the strength and direction of the association between independent and dependent variables. A p-value of less than 0.05 was considered statistically significant. The results of the analyses were presented using text, tables, and charts for clarity.

### Ethical consideration

Ethical clearance was obtained from the Institutional Review Board of Jimma University Institute of Health, permission from the zonal health department, and written informed consent from all participants by discussing the aim and potential benefits of the study.

## Results

### Socio-demographic characteristics of respondents

A study of 421CBHI beneficiaries found a response rate of 99.3% (419 respondents), with the average age was 38.69 ± 6.02 years, 55.8% females, 63.2%) rural dwellers, and 58.2% having a wealth of $200 USD (Table 1).

### Satisfaction towards the facility

The majority of respondents (74.7%) were dissatisfied with the lack of a sufficient reception area, while 67.8% were satisfied with entertainment facilities, sitting arrangements, and client restroom cleanliness. However, over half of study participants (54.9%) expressed dissatisfaction with the laboratory service's distance from their residential area (Table 2).

### Satisfaction towards laboratory professionals service provision and ethical characteristics

Most participants were satisfied with service providers' friendly welcome, specimen collection procedures, and professional appearances, but 58.0% were dissatisfied with their explanations during sample collection (Table 3).

**Table 1. Socio-demographic characteristics of the respondents on patient satisfaction with clinical laboratory services received at public hospitals in Jimma zone, Oromia region, Southwest Ethiopia, 2023, (N = 419).**

| Variables | Category | Frequency | Percentage (%) |
|---|---|---|---|
| Gender | Male | 185 | 44.2 |
| | Female | 234 | 55.8 |
| Age | 18–24 years | 49 | 11.7 |
| | 25–34 years | 148 | 35.3 |
| | 35–44 years | 105 | 25.1 |
| | 45–54 years | 98 | 23.4 |
| | 55 and above years | 19 | 4.5 |
| Residence | Urban | 154 | 36.8 |
| | Rural | 265 | 63.2 |
| House-hold wealth index monthly income | 100USD | 88 | 21.0 |
| | 200USD | 244 | 58.2 |
| | 300USD | 57 | 13.6 |
| | 400USD | 30 | 7.2 |
| Educational status | Illiterate | 58 | 13.8 |
| | Read and write | 164 | 39.1 |
| | Primary | 128 | 30.5 |
| | Secondary | 36 | 8.6 |
| | Diploma and above | 33 | 7.9 |
| Language | Afan Oromo | 287 | 68.5 |
| | Amharic | 107 | 25.5 |
| | other languages | 25 | 6.0 |
| Marital status | Single | 88 | 21.0 |
| | Married | 282 | 67.3 |
| | Divorced/widowed | 49 | 11.7 |
| Number of hospital visits | One visits | 85 | 20.3 |
| | Two visits | 99 | 23.6 |
| | More than two visits | 235 | 56.1 |
| Household family size | 1–5 | 177 | 42.2 |
| | 6–10 | 104 | 24.8 |
| | 11–14 | 80 | 19.1 |
| | >15 | 58 | 13.8 |
| Occupation | Farmer | 118 | 28.2 |
| | Merchant | 126 | 30.1 |
| | Daily laborer | 86 | 20.5 |
| | Others | 89 | 21.2 |

## Satisfaction towards availability of tests and service-related issues

The study found that 70% of participants were satisfied with laboratory test availability, information provision, and service continuity, but higher dissatisfaction rates were observed in the availability of requested tests, CBHI user ID card receiving time, and waiting time (Table 4).

## Overall satisfaction level of CBHI beneficiaries

The mean satisfaction level among beneficiaries of CBHI in clinical laboratory services was 3.43±0.6. The Likert scale rating indicated that a majority of respondents (57.5%) were satisfied with the clinical laboratory services provided by selected hospitals (See Fig 1).

**Table 2. Satisfaction level of the study participants towards facility related variables at public hospitals in Jimma zone, Oromia region, Southwest Ethiopia, 2023, (N = 419).**

| S. no | Variables | Dissatisfied | Neutral | Satisfied | Mean |
|---|---|---|---|---|---|
| | | N (%) | N (%) | N (%) | |
| 1 | Availability of sufficient reception area | 313(74.7) | 52(12.4) | 54(12.9) | 2.19 |
| 2 | Availability of entertaining materials at the lab result waiting area | 82(19.6) | 53(12.6) | 284(67.8) | 3.92 |
| 3 | Availability of benches or chairs sitting arrangements in waiting area | 140(33.5) | 18(4.3) | 261(62.3) | 3.51 |
| 4 | Comfortable with the laboratory setup or organizational structure | 92(22.0) | 197(47) | 130(31.0) | 3.12 |
| 5 | Cleanliness of the laboratory rooms | 151(36.1) | 43(10.3) | 225(53.7) | 3.28 |
| 6 | Cleanness and comfort of the latrine | 122(29.1) | 42(10) | 254(60.6) | 3.50 |
| 7 | Distance from residential area to nearby health care facility | 230(54.9) | 92(22) | 63(23.2) | 3.37 |

**Table 3. Satisfaction level of the respondents towards lab professionals attitude and behaviour at public hospitals in Jimma zone, Oromia region, Southwest Ethiopia, 2023, (N = 419).**

| S. no | Variables | Dissatisfied | Neutral | Satisfied | Mean |
|---|---|---|---|---|---|
| | | N (%) | N (%) | N (%) | |
| 1 | Services providers friendly how they well come you | 53(12.6) | 15(3.6) | 351(83.8) | 4.39 |
| 2 | Procedures for blood and other body fluid specimen collection | 51(12.2) | 1(0.2) | 367(87.6) | 4.54 |
| 3 | Maintenance of patients' privacy and confidentiality in laboratory rooms | 90(21.4) | 22(5.3) | 307(73.3) | 3.92 |
| 4 | The ability of service provider explanation about diagnostic test during sample collection | 222(58.0) | 80(19.1) | 119(27.9) | 2.72 |
| 5 | Availability of laboratory staff on working hours | 51(12.2) | 62(14.8) | 306(73.0) | 3.96 |
| 6 | Laboratory personnel's professional appearances (neatness, professional dressing) | 4(01.0) | 3(0.7) | 412(98.3) | 4.45 |

## Factors associated with satisfaction level of beneficiaries

Six socio-demographic characteristics were analyzed using multiple logistic regressions, with only two showing a significant association with dissatisfaction. High school, illiterate, primary education, and reading and writing patients had a higher likelihood of dissatisfaction compared to those with a diploma and above. Patients who visited the hospital more than two times also had a higher likelihood of dissatisfaction (Table 5).

## Discussion

This study investigated patient satisfaction with clinical laboratory services among CBHI beneficiaries in public hospitals in Jimma Zone, Ethiopia. Here, we discuss the key findings, their alignment with existing literature, and potential explanations for observed patterns. Currently,

**Table 4. Satisfaction towards availability of tests and service-related issues at public hospitals in Jimma zone, Oromia region, Southwest Ethiopia, 2023, (N = 419).**

| S.no | Variables | Dissatisfied | Neutral | Satisfied | Mean |
|---|---|---|---|---|---|
| | | N (%) | N (%) | N (%) | |
| 1 | The availability of the requested laboratory tests | 241 (57.5) | 80(19.1) | 98(23.4) | 2.68 |
| 2 | Availability of proper and clear direction of each lab rooms/sections | 231(55.1) | 33(7.9) | 155(37.0) | 2.98 |
| 3 | Status of happiness with laboratory opening time | 85(20.3) | 28(6.7) | 306(70.0) | 3.96 |
| 4 | Availability of sufficient information provision | 118(28.1) | 37(8.8) | 264(63.1) | 3.75 |
| 5 | Duration of waiting time for lab test result collection | 321(76.6) | 31(7.4) | 67(16.0) | 2.24 |
| 6 | Duration of CBHI users ID receiving time | 255(60.9) | 35(8.4) | 129(30.8) | 2.56 |
| 7 | Laboratory service continuity | 122(15.8) | 42(10) | 255(60.9) | 3.47 |

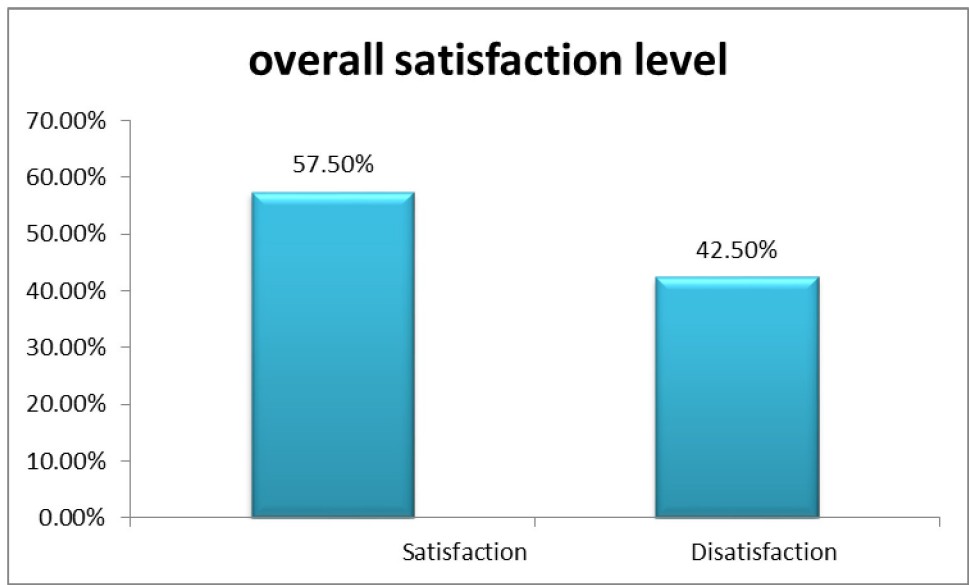

**Fig 1. Overall satisfaction level of the respondents on patients satisfaction with clinical laboratory services received at public hospitals in Jimma Zone, Oromia Region, Southwest Ethiopia, 2023, (N = 419).**

establishing community-based health insurance has been getting high attention, especially in resource-limited countries like Ethiopia, to improve health care utilization and ensure financial protection for households to mitigate poverty [17].

The overall satisfaction level in our study (57.5%) aligns with some previous research in Ethiopia (52.6–60.4%) [49–54]. However, it falls short of findings from other studies reporting satisfaction as high as 90.8% [55, 56]. These variations might stem from differences in patient loads, healthcare infrastructure, and CBHI beneficiary sizes, which can vary across locations. Differences in demographics, expectations, and prior experiences can influence satisfaction levels. Variations in sample size, sampling techniques, and study design can affect results.

This study found a lower satisfaction level compared to a systematic review conducted in Ethiopia, which reported a pooled satisfaction level of 66.0% [57]. This discrepancy might be due to several factors, including differences in sample size, sampling procedures, and the inherent nature of a systematic review, which analyzes data from multiple studies across regions.

The mean satisfaction score of community-based health insurance beneficiaries with clinical laboratory services was 3.43 ± 0.60, ranging from 2.19 to 4.54. This score aligns with findings from other studies conducted in eastern Ethiopia [55]. However, our results were higher compared to another study conducted in Addis Ababa [49]. This variation might be attributed to differences in patient load, lifestyle factors, the condition of healthcare facilities, and service delivery standards between the two study populations.

Beneficiaries expressed positive perceptions regarding staff availability (3.96), professional appearance (4.45), lab opening hours (3.96), and blood sample collection procedures (4.54). This aligns with other studies [55, 58].

A study in Addis Ababa similarly found the lowest satisfaction scores for laboratory professionals' explanation of tests [49]. In contrast, a Northeast Ethiopian study reported a mean satisfaction score of 3.8 out of 5 for clinicians using laboratory services, with praise for staff availability and turnaround time.

**Table 5. Multivariate analysis showing factors associated with satisfaction level of the respondents on patient satisfaction with clinical laboratory services received at public hospitals in Jimma zone, Oromia region, Southwest Ethiopia, 2023, (N = 419).**

| Variables | Category | Satisfaction level | | COR | AOR | 95% CI for AOR | | P-value |
|---|---|---|---|---|---|---|---|---|
| | | Satisfied | Dissatisfied | | | Lower bound | higher bound | |
| Households wealth index | 100USD | 52(12.4) | 36(8.6) | 1.9 | 1.8 | 0.7 | 4.9 | 0.22 |
| | 200USD | 135(32.2) | 109(26) | 2.2 | 2.1 | 0.8 | 5.2 | 0.11 |
| | 300USD | 32(7.6) | 25(6) | 2.1 | 2.1 | 0.7 | 6.0 | 0.17 |
| | 400USD | 22(5.3) | 8(1.9) | 1 | 1 | | | |
| Occupation | Farmer | 76(18.1) | 42(10) | 1 | 1 | | | |
| | Merchant | 81(19.3) | 45(10.7) | 0.5 | 0.7 | 0.4 | 1.4 | 0.36 |
| | daily laborer | 41(9.8) | 45(10.7) | 1.03 | 1.05 | 0.6 | 2.0 | 0.88 |
| | Others | 43(10.3) | 46(11) | 2.0 | 1.7 | 1.0 | 3.1 | 0.16 |
| Educational status | Illiterate | 21(6.4) | 31(7.4) | 8.3 | 6.8 | 2.0 | 23.6 | .002* |
| | Read and write | 98(23.4) | 66(15.5) | 4.9 | 4.0 | 1.3 | 13.0 | .019* |
| | Primary | 71(16.9) | 57(13.6) | 5.8 | 4.6 | 1.4 | 15.0 | .012* |
| | Secondary | 16(3.8) | 20(4.8) | 9.1 | 7.8 | 2.1 | 28.6 | .002* |
| | College + | 29(6.9) | 4(1) | 1 | 1 | | | |
| Language | Afan Oromo | 171(40.8) | 116(27.7) | 1 | 1 | | | |
| | Amharic | 55(13.1) | 52(12.4) | 1.4 | 1.3 | 0.8 | 2.2 | 0.24 |
| | Others | 15(3.6) | 10(2.4) | 0.98 | 0.9 | 0.4 | 2.4 | 0.90 |
| Number of hospital visits | One visits | 56(13.4) | 29(6.9) | 1 | | | | |
| | Two visits | 65(15.5) | 34(8.1) | 1.01 | 0.9 | 0.5 | 1.9 | 0.86 |
| | Above two visits | 120(28.6) | 115(27.4) | 1.9 | 1.7 | 0.98 | 3.1 | 0.049 |
| Household family size | 1–5 | 99(23.6) | 78(18.6) | 1 | 1 | | | |
| | 6–10 | 67(16) | 37(8.8) | 0.7 | 0.7 | 0.4 | 1.2 | 0.20 |
| | 11–15 | 38(9.1) | 42(10) | 1.4 | 1.4 | 0.8 | 2.5 | 0.20 |
| | >15 | 37(8.8) | 21(5) | 0.7 | 0.8 | 0.4 | 1.5 | 0.48 |

Key; *statically significant, (p < 0.05), COR, crude odds ratio, AOR adjusted odds ratio, 1 reference category, Max, VIF = 1.06(no multicollinearity: at VIF < 5). The overall p-value for number of hospital visits and educational status is <0.05

Beneficiaries reported dissatisfaction with waiting times for test results (2.24), similar to findings in eastern Ethiopia [59]. Explanations of test procedures by staff received low satisfaction scores (2.72), echoing concerns raised in prior research [49, 51].

The study identified educational status and the number of hospital visits as significantly associated with satisfaction levels. People with lower education levels are more likely to be dissatisfied with the service compared to those with college degrees. People who visit the hospital more than twice are more likely to be dissatisfied compared to those who visit only once. This aligns with previous studies in Ethiopia [49, 52]. However, factors like age, sex, and residence did not show significant associations, similar to findings from Addis Ababa [49].

## Limitation of the study

The study evaluated patient satisfaction with clinical laboratory services in community-based health insurance, excluding other services. Hospital-based interviews may result in social desirability bias, as patients may favor healthcare providers. This study employed a cross-sectional design, which limits the ability to establish causal relationships between factors and patient satisfaction. Since data collection occurred at a single point in time, we cannot determine if factors identified through the study caused variations in satisfaction levels. A

convenient sampling technique was used to select study participants within the chosen hospitals. While this method allowed for efficient data collection, it might introduce selection bias as participants who were readily available may not be fully representative of the entire CBHI beneficiary population that utilizes laboratory services within these hospitals. The study focused on CBHI beneficiaries in public hospitals located within the Jimma Zone of Ethiopia. This specific geographic focus may limit the generalizability of the findings to other populations or healthcare settings.

## Conclusion

The overall satisfaction level of CBHI beneficiaries with clinical laboratory services at selected public hospitals in Jimma Zone was moderate. While beneficiaries appreciated the professionalism of laboratory personnel and specimen collection procedures, dissatisfaction arose due to limited reception space and extended waiting times for results. Notably, educational background and frequency of hospital visits influenced satisfaction levels. Expanding reception areas can alleviate overcrowding and create a more comfortable waiting environment. Additionally, optimizing workflows can reduce wait times for test results, improving overall efficiency. Implementing routine explanations of test purposes and procedures during specimen collection, regardless of patient education level, can foster a sense of trust and informed participation in their healthcare journey.

## Supporting information

**S1 File. Survey questions in the English and sampling procedure.**
(DOCX)

**S1 Dataset.**
(DTA)

## Acknowledgments

We would like to thank the study participants, data collectors and those who had contribution to this study.

## Author Contributions

**Conceptualization:** Mujahid Girma, Zewudineh Sahilemariam, Temesgen Kabeta.

**Data curation:** Nigusu Getachew, Zewudineh Sahilemariam, Amit Arora.

**Formal analysis:** Nigusu Getachew, Mujahid Girma, Temesgen Kabeta.

**Methodology:** Zewudineh Sahilemariam, Temesgen Kabeta, Amit Arora.

**Supervision:** Nigusu Getachew, Mujahid Girma, Amit Arora.

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
