## [Decision Letter · Decision Letter 0]

28 May 2024

PONE-D-24-14840Community-Based Health Insurance Beneficiaries satisfaction on Laboratory Services and Associated Factors in Selected Public Hospitals in Jimma Zone, Oromia Region, Southwest EthiopiaPLOS ONE

Dear Dr. Endashaw,

Thank you for submitting your manuscript to PLOS ONE. After careful consideration, we feel that it has merit but does not fully meet PLOS ONE’s publication criteria as it currently stands. Therefore, we invite you to submit a revised version of the manuscript that addresses the points raised during the review process.

We look forward to receiving your revised manuscript.

Kind regards,

Kiddus Yitbarek, MPH

Academic Editor

PLOS ONE

“Funding: This work was supported by Jimma University”

3. We note that your Data Availability Statement is currently as follows: [All relevant data are within the manuscript and its Supporting Information files]

5. We are unable to open your Supporting Information file [CBHI spss.sav]. Please kindly revise as necessary and re-upload.

Additional Editor Comments:

Dear Authors,

Thank you for your interesting research on healthcare delivery in Ethiopia, focusing on the perspective of health insurance policyholders regarding laboratory services. Your study highlights a previously overlooked aspect, and we appreciate the contribution.

However, there are points that require further elaboration to strengthen the research:

1. Rationale:

You rightly point out the importance of client satisfaction for healthcare system acceptance. In your study design, you focused on a specific group, CBHI members. Could you elaborate on the rationale for selecting this specific population in the background section?

2. Measurement Reliability:

Another crucial aspect is the reliability of the satisfaction measurement instruments. Would you be able to report on the reliability of the specific items used to assess satisfaction?

3. Satisfaction Measurement Method:

You've described how satisfaction was measured using a five-point Likert scale and the construction of the composite measure. Could you compare your chosen method (mean scores) for determining a satisfaction/dissatisfaction reference point with other commonly used methods?

4. Measurement of Other Variables:

The methods section doesn't detail how other variables, such as the wealth index, were measured. Could you elaborate on the specific procedures used to measure wealth and any other relevant variables in your study?

5. Reference Category in Regression Model:

Your regression model utilizes "Others" as the reference category. This might not convey the most informative message. Could you consider revising this aspect of the model?

The reviewers have raised valuable points that can significantly enhance your research quality. We encourage you to address each comment point-by-point in your revised manuscript and provide explanations for the feedback received

Reviewers' comments:

Reviewer's Responses to Questions

**Comments to the Author**

1. Is the manuscript technically sound, and do the data support the conclusions?

Reviewer #1: Yes

Reviewer #2: Partly

2. Has the statistical analysis been performed appropriately and rigorously? 

Reviewer #1: Yes

Reviewer #2: Yes

3. Have the authors made all data underlying the findings in their manuscript fully available?

Reviewer #1: Yes

Reviewer #2: Yes

4. Is the manuscript presented in an intelligible fashion and written in standard English?

Reviewer #1: No

Reviewer #2: No

5. Review Comments to the Author

Reviewer #1: This study provides valuable insights into the satisfaction levels of Community-Based Health Insurance beneficiaries regarding laboratory services in Jimma Zone, Ethiopia. The authors have conducted a comprehensive assessment, utilizing a facility-based cross-sectional design and employing rigorous statistical analyses to explore various factors associated with satisfaction. The findings shed light on both areas of strength and areas for improvement within the healthcare system, particularly in public hospitals.

However, to improve the output, I have listed my comments and suggestions.

General recommendations:

• Line numbers are recommended for the reviewer to detect and label feedback.

• Spacing and indentation need attention (e.g., Introduction paragraph 4; Client …..).

• English grammar should be carefully checked.

Title: Cases should be appropriately oriented. Updated title “Community-Based Health Insurance Beneficiaries Satisfaction on Laboratory Services and Associated Factors in Selected Public Hospitals in Jimma Zone, Oromia Region, Southwest, Ethiopia”

Abstract:

• Words in the first text should be written in full. For example, "SPSS" should be written as "Statistical Package of Social Sciences version 25."

• It is recommended to include the distribution of participants' ages in the summary.

Introduction:

• Authors should consistently emphasize abbreviations throughout the paper, such as WHO, OOPS, NGOs, LMICs, CBHI. Some abbreviations are repeatedly used with the full term while others are not. Please ensure consistency in their usage throughout the paper.

• Caution and revision are needed in citations. It's not recommended to include a bunch of references per paragraph. For example, in the second paragraph, 16 articles were cited at once (9-24), which could be overly vague. Instead, it's recommended to cite specific articles at each instance.

Methodology

• In the Study Design section, it is recommended to include the statement "Fifty percent (50%) of the hospitals were randomly selected for community-based health insurance. That means about three of them (50%) were selected randomly by the lottery method." under the sample size determination.

• The Population section should be separated from the concept of eligibility criteria and placed in its own section. For example: "The study included CBHI beneficiaries who attended selected hospitals for laboratory services during the study period, while those who were unable to hear or speak were excluded." Additionally, clarification is needed regarding the rationale for excluding individuals unable to hear or speak, particularly since they were already at the healthcare facility. Was their exclusion due to severe illness affecting consent or other justifications?

• In the Abstract section, mention of bivariate and multivariable analysis should also be included in the main section for consistency.

• While not mandatory, I personally recommend additional analysis to evaluate the goodness of model fit using the Hosmer-Lemeshow test, along with reporting the associated p-value. Additionally, it would be beneficial to assess multicollinearity using variance inflation factors (VIFs)."

Result

• How was the household wealth index scaled in your study?

Discussion

I would like to address the beginning of the Discussion section, which I found lacking in clarity and engagement. Here are some suggestions for improvement:

• The authors should interpret their findings in the context of existing literature, acknowledge limitations, and provide insights for future research and practical implications.

• Begin by summarizing the key findings of the study, highlighting both areas of strength and areas for improvement in CBHI beneficiaries' satisfaction with laboratory services. Discuss how these findings align with or diverge from previous research on patient satisfaction in healthcare settings.

• Provide a comprehensive review of relevant literature on patient satisfaction, particularly in the context of CBHI schemes or similar healthcare initiatives in Ethiopia or other comparable settings. Compare the findings of the current study with previous research, noting similarities, differences, and potential explanations for discrepancies.

• Offer explanations for the observed patterns in satisfaction levels and associated factors identified in the study. Consider factors such as socio-demographic characteristics of beneficiaries, healthcare system infrastructure, and quality of service delivery in public hospitals. Discuss how these factors may influence patients' perceptions of laboratory services.

Limitation

• What about as the cross-sectional design, potential biases introduced by the sampling method, and limitations in generalizability due to the study's focus on a specific geographic area?

References

• It is better to minimize the number of references.

Reviewer #2: This paper has explored CBHI user's satisfaction regarding laboratory services in Jimma zone.

In background

Make sure to introduce all abbreviations in the text before using them (for example "OOPS"). Additionally, once you’ve introduced the abbreviation you can just use it in the remainder of the text ( for example CBHI)

Have you exhausted all possible confounders? (disease severity..)

In method:

• Clarify the sampling frame and the sampling method; "focusing on six hospitals: eight governmental, two private and 120 health centers. Fifty percent (50%) of the hospitals were randomly selected for community-based health insurance." This part is a bit confusing on the study population section.

• Any justifications as to why those with visual and auditory impairment/difficulties were excluded?

• Was there any other inclusion or exclusion criterion that was used?

• On table 5 of the results: try to use uniform reference categories (either first or last)

On the discussion

• Consider adding the interpretations of the odds ratios to further elaborate the findings.

• Can we compare studies viewing overall satisfaction with one viewing just laboratory service related?

General concern regarding the interview tool since it's not an internationally validated tool, could you do a validity check (internal consistency..) ?

Overall review the structure of the paragraphs in terms of global, regional and local data as well as in terms of associated factors, refine the grammar and consider getting it proof read.

6. PLOS authors have the option to publish the peer review history of their article (what does this mean?). If published, this will include your full peer review and any attached files.

Reviewer #1: **Yes: **Eyob Girma Abera

Reviewer #2: No

---

## [Author Response · Author response to Decision Letter 0]

20 Jun 2024

Point-by-point response to review questions 

Dear Editor,

Thank you for facilitating the review of our manuscript and the reviewers for their invaluable comments. After careful review of all the invaluable comments raised by the editor and reviewers, we have prepared a point-by-point response and also revised the manuscript accordingly. 

Additional Editor Comments:

Dear Authors,

Thank you for your interesting research on healthcare delivery in Ethiopia, focusing on the perspective of health insurance policyholders regarding laboratory services. Your study highlights a previously overlooked aspect, and we appreciate the contribution.

However, there are points that require further elaboration to strengthen the research:

1. Rationale:

You rightly point out the importance of client satisfaction for healthcare system acceptance. In your study design, you focused on a specific group, CBHI members. Could you elaborate on the rationale for selecting this specific population in the background section?

Response: thank you very much for this important concern. I have corrected it in the revised copy of the article. 

2. Measurement Reliability:

Another crucial aspect is the reliability of the satisfaction measurement instruments. Would you be able to report on the reliability of the specific items used to assess satisfaction?

Response: thank you very much for your insight. I have corrected it in the revised copy of the article. 

3. Satisfaction Measurement Method:

You've described how satisfaction was measured using a five-point Likert scale and the construction of the composite measure. Could you compare your chosen method (mean scores) for determining a satisfaction/dissatisfaction reference point with other commonly used methods?

Response: thank you very much for this important concern. The study described uses the mean score on a Likert scale to determine a satisfaction/dissatisfaction reference point. This is a common method, but it has some limitations. Here's a comparison with other commonly used methods: 1. Mean Score:

Strengths: Easy to calculate, readily interpretable.

Weaknesses: Doesn't account for the distribution of responses. A high mean score could be due to everyone being slightly satisfied, or a few very satisfied people dragging the average up.

Good for: Getting a general sense of satisfaction, especially if the data is normally distributed.

2. Median Score:

Strengths: Less sensitive to outliers than the mean, reflects the "middle ground" of responses.

Weaknesses: Doesn't use all the data available, less intuitive than the mean for some audiences.

Good for: When there might be extreme scores that skew the data set.

3. Percentage:

Strengths: Shows the distribution of responses, provides a more detailed picture of satisfaction levels.

Weaknesses: Can be overwhelming with many categories, less easy to summarize in a single number.

Good for: When a detailed breakdown of satisfaction levels is needed across different categories.

The mean score is a common and easy-to-understand method but based on the specific research question and data we have the option to see other method. 

4. Measurement of Other Variables:

The methods section doesn't detail how other variables, such as the wealth index, were measured. Could you elaborate on the specific procedures used to measure wealth and any other relevant variables in your study?

Response: Thank you for your concern regarding the household wealth index. While the table shows income categories, our wealth index is primarily based on household assets such as land, cash crops, and other resources. Income can be a factor influencing wealth, but this approach provides a more comprehensive picture of socioeconomic status.

5. Reference Category in Regression Model:

Your regression model utilizes "Others" as the reference category. This might not convey the most informative message. Could you consider revising this aspect of the model?

Response: thank you very much for this important concern. I have corrected it in the revised copy of the article. 

Reviewer #1: Comments

This study provides valuable insights into the satisfaction levels of Community-Based Health Insurance beneficiaries regarding laboratory services in Jimma Zone, Ethiopia. The authors have conducted a comprehensive assessment, utilizing a facility-based cross-sectional design and employing rigorous statistical analyses to explore various factors associated with satisfaction. The findings shed light on both areas of strength and areas for improvement within the healthcare system, particularly in public hospitals.

However, to improve the output, I have listed my comments and suggestions.

General recommendations:

• Line numbers are recommended for the reviewer to detect and label feedback.

• Spacing and indentation need attention (e.g., Introduction paragraph 4; Client …..).

• English grammar should be carefully checked.

Title: Cases should be appropriately oriented. Updated title “Community-Based Health Insurance Beneficiaries Satisfaction on Laboratory Services and Associated Factors in Selected Public Hospitals in Jimma Zone, Oromia Region, Southwest, Ethiopia”

Response: thank you very much for this important concern. I have corrected it in the revised copy of the article. 

Abstract:

• Words in the first text should be written in full. For example, "SPSS" should be written as "Statistical Package of Social Sciences version 25."

• It is recommended to include the distribution of participants' ages in the summary.

Response: thank you, corrected it in the revised copy.

Introduction:

• Authors should consistently emphasize abbreviations throughout the paper, such as WHO, OOPS, NGOs, LMICs, CBHI. Some abbreviations are repeatedly used with the full term while others are not. Please ensure consistency in their usage throughout the paper.

Response: thank you so much for your insight; we have corrected it in the revised copy.

• Caution and revision are needed in citations. It's not recommended to include a bunch of references per paragraph. For example, in the second paragraph, 16 articles were cited at once (9-24), which could be overly vague. Instead, it's recommended to cite specific articles at each instance.

Response: thank you so much for your insight; we have corrected it in the revised copy of the article accordingly.

• In the Study Design section, it is recommended to include the statement "Fifty percent (50%) of the hospitals were randomly selected for community-based health insurance. That means about three of them (50%) were selected randomly by the lottery method." under the sample size determination

Response: thank you so much for your insight; we have corrected it in the revised copy of the article accordingly.

• The Population section should be separated from the concept of eligibility criteria and placed in its own section. For example: "The study included CBHI beneficiaries who attended selected hospitals for laboratory services during the study period, while those who were unable to hear or speak were excluded." Additionally, clarification is needed regarding the rationale for excluding individuals unable to hear or speak, particularly since they were already at the healthcare facility. Was their exclusion due to severe illness affecting consent or other justifications?

Response: thank you for your concern: Eligibility Criteria: CBHI beneficiaries who met the following criteria were included in the study: Attended one of the selected hospitals for laboratory services during the study period and able to provide informed consent were included 

Exclusion Criteria: Individuals who were unable to hear or speak were excluded from the study.

Justification for Exclusion: Individuals who were unable to provide informed consent were excluded due to the study's requirement to understand the research procedures and potential risks and benefits.

Individuals who were unable to hear or speak were excluded due to potential challenges in effectively communicating study information and ensuring comprehension.

• In the Abstract section, mention of bivariate and multivariable analysis should also be included in the main section for consistency.

Response: we agree with your observation, thank you. We have corrected the editorial problem accordingly in the revised copy of the article

• While not mandatory, I personally recommend additional analysis to evaluate the goodness of model fit using the Hosmer-Lemeshow test, along with reporting the associated p-value. Additionally, it would be beneficial to assess multicollinearity using variance inflation factors (VIFs)."

Response: thank you so much for your insight; we have corrected it in the revised copy of the article accordingly.

Result

• How was the household wealth index scaled in your study?

Response: Thank you for your concern regarding the household wealth index. While the table shows income categories, our wealth index is primarily based on household assets such as land, cash crops, and other resources. Income can be a factor influencing wealth, but this approach provides a more comprehensive picture of socioeconomic status.

Discussion

I would like to address the beginning of the Discussion section, which I found lacking in clarity and engagement. Here are some suggestions for improvement:

• The authors should interpret their findings in the context of existing literature, acknowledge limitations, and provide insights for future research and practical implications.

Response: thank you so much for your insight; we have corrected it in the revised copy of the article accordingly.

• Begin by summarizing the key findings of the study, highlighting both areas of strength and areas for improvement in CBHI beneficiaries' satisfaction with laboratory services. Discuss how these findings align with or diverge from previous research on patient satisfaction in healthcare settings.

Response: thank you so much for your insight; we have corrected it in the revised copy of the article accordingly.

• Provide a comprehensive review of relevant literature on patient satisfaction, particularly in the context of CBHI schemes or similar healthcare initiatives in Ethiopia or other comparable settings. Compare the findings of the current study with previous research, noting similarities, differences, and potential explanations for discrepancies.

Response: thank you so much for your insight; we have corrected it in the revised copy of the article accordingly.

• Offer explanations for the observed patterns in satisfaction levels and associated factors identified in the study. Consider factors such as socio-demographic characteristics of beneficiaries, healthcare system infrastructure, and quality of service delivery in public hospitals. Discuss how these factors may influence patients' perceptions of laboratory services.

Response: thank you so much for your insight; we have corrected it in the revised copy of the article accordingly.

Limitation

• What about as the cross-sectional design, potential biases introduced by the sampling method, and limitations in generalizability due to the study's focus on a specific geographic area?

Response: thank you very much; we have corrected it in the revised copy. 

References

• It is better to minimize the number of references. 

Response: thank you; we have corrected it in the revised copy accordingly. 

Reviewer #2: Comments

This paper has explored CBHI user's satisfaction regarding laboratory services in Jimma zone.

In background

Make sure to introduce all abbreviations in the text before using them (for example "OOPS"). Additionally, once you’ve introduced the abbreviation you can just use it in the remainder of the text (for example CBHI) Have you exhausted all possible confounders? (Disease severity...)

Response: thanks a lot; we have corrected it in the revised copy of the article. 

In method:

• Clarify the sampling frame and the sampling method; "focusing on six hospitals: eight governmental, two private and 120 health centers. Fifty percent (50%) of the hospitals were randomly selected for community-based health insurance." This part is a bit confusing on the study population section.

Response: thank you very much for your observation

This study focuses on Public hospitals in the Zone. The sampling frame included a complete list of all functional public (n=6) hospitals only that operating in the area. Fifty percent (50%) of the hospitals were randomly selected for further study into community-based health insurance programs. This resulted in three public hospitals being included in the study.

• Any justifications as to why those with visual and auditory impairment/difficulties were excluded?

• Was there any other inclusion or exclusion criterion that was used?

Response: thank you for your concern: Eligibility Criteria: CBHI beneficiaries who met the following criteria were included in the study: Attended one of the selected hospitals for laboratory services during the study period and able to provide informed consent were included 

Exclusion Criteria: Individuals who were unable to hear or speak were excluded from the study.

Justification for Exclusion: Individuals who were unable to provide informed consent were excluded due to the study's requirement to understand the research procedures and potential risks and benefits.

Individuals who were unable to hear or speak were excluded due to potential challenges in effectively communicating study information and ensuring comprehension.

• On table 5 of the results: try to use uniform reference categories (either first or last)

Response: thank you very much for your insight: The reference category can be chosen based on the research question or the variable itself. In the table, the author likely chose College+ in education as the reference because it might represent the highest level of education achieved and allows for easier comparison with other categories. We are not focusing on uniformity of the reference categories 

On the discussion

• Consider adding the interpretations of the odds ratios to further elaborate the findings.

• Can we compare studies viewing overall satisfaction with one viewing just laboratory service related?

Response: thank you very much; we have corrected it in the revised copy of the article. 

General concern regarding the interview tool since it's not an internationally validated tool, could you do a validity check (internal consistency..) ?

Response: thank you very much for you observation; we have corrected it in the revised copy of the article

Overall review the structure of the paragraphs in terms of global, regional and local data as well as in terms of associated factors, refine the grammar and consider getting it proof read.

Response: thank you very much; we have corrected it in the revised copy of the article.

---

## [Decision Letter · Decision Letter 1]

22 Jul 2024

PONE-D-24-14840R1Community-Based Health Insurance Beneficiaries satisfaction on Laboratory Services and Associated Factors in Selected Public Hospitals in Jimma Zone, Oromia Region, Southwest EthiopiaPLOS ONE

Dear Dr. Getachew,

Thank you for submitting your manuscript to PLOS ONE. After careful consideration, we feel that it has merit but does not fully meet PLOS ONE’s publication criteria as it currently stands. Therefore, we invite you to submit a revised version of the manuscript that addresses the points raised during the review process.

Dear Dr. Getachew,

As I can see from the revised version of your manuscript, many improvements have been made. However, there are still issues with grammar, writing style, and formatting that need to be addressed. I strongly recommend that a copy editor review the entire manuscript before publication.

We look forward to receiving your revised manuscript.

Kind regards,

Kiddus Yitbarek, MPH

Academic Editor

PLOS ONE

Journal Requirements:

Reviewers' comments:

Reviewer's Responses to Questions

**Comments to the Author**

1. If the authors have adequately addressed your comments raised in a previous round of review and you feel that this manuscript is now acceptable for publication, you may indicate that here to bypass the “Comments to the Author” section, enter your conflict of interest statement in the “Confidential to Editor” section, and submit your "Accept" recommendation.

Reviewer #1: All comments have been addressed

Reviewer #2: All comments have been addressed

2. Is the manuscript technically sound, and do the data support the conclusions?

Reviewer #1: Yes

Reviewer #2: Yes

3. Has the statistical analysis been performed appropriately and rigorously? 

Reviewer #1: Yes

Reviewer #2: Yes

4. Have the authors made all data underlying the findings in their manuscript fully available?

Reviewer #1: Yes

Reviewer #2: Yes

5. Is the manuscript presented in an intelligible fashion and written in standard English?

Reviewer #1: Yes

Reviewer #2: Yes

6. Review Comments to the Author

Reviewer #1: I personI am satisfied with the responses and want to give credit for the great work. I would only like to recommend the authors recheck the reference format, e.g., 4, 7, & 8.

Reviewer #2: Most of the comments have been addressed,but I would recommend you either rephrase or revise the eligibility criteria

7. PLOS authors have the option to publish the peer review history of their article (what does this mean?). If published, this will include your full peer review and any attached files.

Reviewer #1: **Yes: **Eyob Girma Abera

Reviewer #2: No

---

## [Author Response · Author response to Decision Letter 1]

27 Jul 2024

Point-by-point response to review questions 

Dear Editor,

Thank you for facilitating the review of our manuscript and the reviewers for their invaluable comments. After careful review of all the invaluable comments raised by the editor and reviewers, we have prepared a point-by-point response and also revised the manuscript accordingly. 

Additional Editor Comments:

Reviewer #1: 

I person I am satisfied with the responses and want to give credit for the great work. I would only like to recommend the authors recheck the reference format, e.g., 4, 7, & 8.

Response: thank you very much for this important concern. I have corrected it in the revised copy of the article. 

Reviewer #2: 

Most of the comments have been addressed, but I would recommend you either rephrase or revise the eligibility criteria

Response: thank you very much for this important concern. I have corrected it in the revised copy of the article.

---

## [Editor Report · Decision Letter 2]

31 Jul 2024

Community-Based Health Insurance Beneficiaries satisfaction on Laboratory Services and Associated Factors in Selected Public Hospitals in Jimma Zone, Oromia Region, Southwest Ethiopia

PONE-D-24-14840R2

Dear Dr. Getachew,

We’re pleased to inform you that your manuscript has been judged scientifically suitable for publication and will be formally accepted for publication once it meets all outstanding technical requirements.

Kind regards,

Kiddus Yitbarek, MPH

Academic Editor

PLOS ONE
---

## [Editor Report · Acceptance letter]

5 Aug 2024

PONE-D-24-14840R2 

PLOS ONE

Dear Dr. Getachew, 

I'm pleased to inform you that your manuscript has been deemed suitable for publication in PLOS ONE. Congratulations! Your manuscript is now being handed over to our production team.

Kind regards, 

on behalf of

Mr. Kiddus Yitbarek 

Academic Editor

PLOS ONE